# Adaptive Designs: Lessons for Inflammatory Bowel Disease Trials

**DOI:** 10.3390/jcm9082350

**Published:** 2020-07-23

**Authors:** Ferdinando D’Amico, Silvio Danese, Laurent Peyrin-Biroulet

**Affiliations:** 1Department of Biomedical Sciences, Humanitas University, Pieve Emanuele, 20090 Milan, Italy; damico_ferdinando@libero.it (F.D.); sdanese@hotmail.com (S.D.); 2Department of Gastroenterology and Inserm NGERE U1256, Nancy University Hospital, University of Lorraine, 54500 Vandoeuvre-lès-Nancy, France; 3Department of Gastroenterology, Humanitas Clinical and Research Center-IRCCS, Rozzano, 20089 Milan, Italy

**Keywords:** adaptive designs, study design, inflammatory bowel disease

## Abstract

In recent decades, scientific research has considerably evolved in the field of inflammatory bowel diseases (IBD) and clinical studies have become increasingly complex, including new outcomes, different study populations, and additional techniques of re-randomization and centralized control. In this context, randomized clinical trials are the gold standard for new drugs’ development. However, traditional study designs are time-consuming, expensive, and only a small percentage of the tested therapies are approved. For this reason, a new study design called “adaptive design” has been introduced, allowing to accumulate data during the study and to make predefined adjustments based on the results of scheduled interim analysis. Our aim is to clarify the advantages and drawbacks of adaptive designs in order to properly interpret study results and to identify their role in upcoming IBD trials.

## 1. Introduction

Randomized clinical trials are considered the gold standard for clinical research and have enormously contributed to the development of new drugs and to the advancement of scientific knowledge [1]. However, clinical trial phases of a new drug are very long and expensive, requiring more than 10 years from the research start until drug approval [2]. Furthermore, clinical trials are becoming more and more complex and ever larger sample sizes are needed to achieve adequate statistical power, making patient recruitment difficult [1]. These limitations are evident in the field of inflammatory bowel diseases (IBD), as several new molecules have been tested since the biological drugs’ era began [3]. In fact, the molecules must pass a clinical trial phase in vitro or on animals and then be tested on humans [2]. Firstly, their safety and their pharmacokinetic and pharmacodynamic profiles have to be investigated (phase I), and secondly their efficacy has to be evaluated, identifying the optimal dose drug (phase II) [2]. If all these steps are passed, the drug can be experimented in large phase III studies aiming to confirm efficacy and evaluate safety, assessing the onset of rare adverse events, which would not be found in small sample investigations [2]. Finally, once the drug has been approved, pharmacovigilance studies (phase 4) are conducted on thousands of people to monitor the drug safety profile [4]. The experimental phase can be interrupted in any of the described phases [2]. Importantly, an apparently safe and effective drug in I and II phases may result as ineffective or dangerous in phase III trials [5,6]. Similarly, severe adverse events may be detected during the post-marketing phase resulting in drug withdrawal [7]. For this reason, as the cost of new drugs’ development ranges from ~USD 3 billion to more than USD 30 billion per approval, it is necessary to optimize resources and to reduce research times [8]. To overcome these limitations, new study designs, called adaptive designs, have been specifically conceived [1]. They use data accumulated during the trial at one or several interim time-points to decide predetermined modifications of the study design without undermining the statistical validity of the results (Figure 1) [9]. According to the modification performed, different adaptive designs can be recognized [1]. The purpose of our review was to summarize literature evidence on adaptive designs, highlighting their advantages and drawbacks in order to identify their role in the upcoming IBD trials.

## 2. Types of Adaptive Designs

Several types of adaptive designs have been described and are currently available [10,11]. Adaptive dose-finding designs are often used in early-phase studies [11]. They consist of multiple study arms and are frequently based on a continual reassessment method [11]. They allow to perform interim analyses of the accumulated data and to modify patients’ allocations to identify the optimal drug dosage to be used in later phases (Table 1) [10,11]. Adaptive hypothesis designs include a pre-planned modification of the study hypothesis (e.g., non-inferiority, superiority) or endpoints (primary and secondary) based on preliminary study results [10]. Other adaptive designs allow to modify dose or treatment duration (adaptive group sequential design), to re-estimate the sample size (sample size re-estimation design), to switch a patient from a treatment group to another due to a loss of response to initial therapy (adaptive treatment-switching design), or to adjust randomization schedules, increasing the number of patients randomized to the most beneficial groups (adaptive randomization design) [10]. In pick-the-winner/drop-the-loser designs, study arms can be modified, removing less effective ones and possibly adding groups that seem effective according to the accumulated data at interim [10]. Other trials can be adapted based on treatment response to biomarkers (biomarker adaptive design) [10]. Moreover, some studies may combine phases II and III of clinical trials in the so-called “seamless phase II/III design” [10,11]. These studies are designed as phase III studies but include an intermediate “phase II” analysis to assess whether to continue or to stop the study [10,11]. Finally, if multiple adaptive designs are combined in the same study, it is called multiple adaptive design [10].

## 3. Advantages and Drawbacks of Adaptive Designs

Advantages and drawbacks of adaptive designs are reported in Figure 2. The adaptive methods must be established before the trials start, setting interim points in which a predetermined committee has the task of evaluating the preliminary results and judging if trial modifications are necessary [12]. It is essential that the interim analysis is blinded and that data are analyzed by an independent committee that has no relationship with the investigators in order to reduce the risk of bias and not to invalidate the interpretation of the results [10]. In fact, the investigators could be influenced in the evaluation of the patient’s condition if, after the interim analysis, they are aware that one treatment is superior to another. Adaptive designs have greater patient and sponsor acceptability than traditional studies [13]. In fact, patients may be more encouraged to accept a study in which randomization can be modified, allowing them to be treated with the most effective approach [13]. Similarly, even sponsors could have greater motivation to finance adaptive designs, accumulating information not only at the end of the study but also while data are being collected [13]. Another advantage of adaptive designs is their short duration, making them particularly effective in emergency situations [14]. Interestingly, their use was proposed during the Ebola epidemic that occurred in West Africa from 2014 to 2016, suggesting that adaptive designs could allow to identify effective therapies with the minimal allocation of patients [14]. Furthermore, the ethical aspect also has an important role and must be taken into account when choosing the study design [12]. In randomized trials, for example, randomization is justified by the need to avoid patient selection bias and it is ethically accepted because the experimenters do not know which study arm is most effective [12]. For this reason, a study should be interrupted as soon as one treatment appears to be more favorable than the other. The choice of an adaptive design, instead, could provide an ethical validation, as preliminary data could allow to allocate patients to the best treatment arm [12]. In addition, adapting the study design by recalculating the sample size or changing the study arms allows one to avoid conducting underpowered studies and to identify the patients who are likely to respond most to a specific treatment. However, adaptive designs have also some limitations that need to be mentioned [13]. First of all, the planning phase of the study is fundamental and requires great attention, determining an increased delay between study planning and beginning. Secondly, in an adaptive decision-making context, if the interim data analysis is not performed at an appropriate time, the study integrity could be impaired. Thirdly, premature modification could affect the results as some endpoints may need more time to be verified. Fourthly, adaptive designs require specific and very elaborate statistical methods which make it very difficult for the scientific community to accept their results. Fifthly, the adaptations could make difficult the study interpretation when pre- and post-adaptation results are discordant.

## 4. Evolution of Adaptive Designs: the Regulatory Authorities’ Point of View

The main challenge for adaptive designs is to define their use in study planning. To answer this question, the Food and Drug Administration (FDA) in 2010 classified adaptive methods as “well-understood” or “less well-understood” based on a statistical point of view and on the level of regulatory approval experience with the specific design [12]. Well-understood designs (e.g., adaptation of study eligibility criteria, sample size re-estimation, and adaptive group sequential design) have a low risk of inconsistency and bias, enhance the study results, and their use is recommended [12]. Less well-understood adaptive methods (e.g., adaptations for dose selection studies or multiple study designs), on the other hand, are not fully supported by scientific evidence, have relatively little regulatory experience, and can distort the interpretation of study results [12]. This adaptive design classification underlines the important role of the statistician, who should be involved in the early planning phase of the studies, minimizing type I errors (false positives). However, the FDA has recently eliminated this classification as more experience has been gained with adaptive designs and some designs defined as less well-understood are currently used and are “more understood” [15,16,17]. The FDA guidance suggests to use adaptive designs when the primary aim of the study cannot be achieved with traditional study designs [17]. Nonetheless, traditional phase II and phase III clinical trials have an estimated failure rate of 62% and 45%, respectively, and the main cause of drug development attrition is the lack of efficacy and safety [18]. A recent study investigated the evolution of adaptive designs, showing the number of protocols conducted and the impact of the study design on trial success [19]. Up to April 2016, 59 adaptive protocols for new drugs’ development had been submitted to the European Medicines Agency (EMA) scientific advice, of which about half started and only 23 ended. Oncology and cardiology were the two branches that mostly adopted this new design. The primary end-point was achieved in nine trials (39%), two trials (9%) were stopped for futility, and four trials (17%) led to marketing drug approval [19]. Dose selection, sample size re-assessment, and stopping for futility were the most common adaptive designs, while primary analysis adaptation, adaptive randomization, and population enrichment were less frequent [19]. On the other hand, a systematic review including studies published between 2012 and 2015 revealed that the sequential design was the most frequently used adaptive design (90.6%), followed by adaptive dose/treatment group selection (8.6%) and adjustments of the sample size (7.8%) [20].

## 5. Lessons from Non-IBD Trials

A randomized, placebo-controlled, double blind trial with seamless phase II/III design assessed the long-term efficacy and safety of oral propranolol for treatment of infantile hemangiomas [21]. Patients were randomized to receive placebo or four different doses of propranolol (1 mg/kg/day for three months, 3 mg/kg/day for three months, 1 mg/kg/day for six months, or 3 mg/kg/day for six months). At the pre-planned interim analysis, a higher rate of therapeutic success was achieved by patients treated with propranolol 3 mg/kg/day for six months compared with the other groups (63% vs. 8% (placebo), 10% (1 mg/kg/day for three months), 8% (3 mg/kg/day for three months), and 38% (1 mg/kg/day for six months), respectively) and the monitoring committee decided to use this drug dosage to evaluate the superiority of propranolol over the placebo, allowing to select the drug dose with the greatest benefit for patients and to reduce the study duration. Importantly, patients treated with propranolol obtained a higher success rate than the placebo at week 24 (60% vs. 4%, *p* < 0.001) confirming drug superiority. Rugo and colleagues conducted a randomized controlled trial with a biomarker-adaptive design to assess the efficacy of different chemotherapy regimens for breast cancer treatment [22]. Ten biomarkers were used to establish patient randomization and to evaluate response to therapy. Drugs with poor response to therapy (<85%) were discontinued for futility allowing not only to administer the drugs with greater probability of success but also to reduce the number of patients undergoing ineffective treatments and to validate the use of biomarkers. Veliparib–carboplatin reached a predicted success probability of 88% and was shown to be more effective than the placebo (51% vs. 26%) in determining the pathological complete response in triple-negative patients (human epidermal growth factor receptor 2 [HER2]–negative, estrogen-receptor–negative, and progesterone-receptor–negative). A phase 4 study with an adaptive hypothesis design evaluated cardiovascular safety and efficacy of saxagliptin in patients with type 2 diabetes mellitus with a history of cardiovascular disease or multiple risk factors for vascular disease [23]. The study aimed to demonstrate the superiority of the experimental drug over the placebo and included an interim analysis for a possible adaptation of the hypothesis (non-inferiority vs. superiority). At the scheduled evaluation, saxagliptin did not meet the superiority criteria and it was therefore decided to test the non-inferiority hypothesis. Importantly, lower levels of glycated hemoglobin were found in the saxagliptin group compared with the placebo at the end of the treatment period (7.7% vs. 7.9%, *p* < 0.001) and no difference in the rate of ischemic events was identified (7.3% vs. 7.2%, *p* <0.001), confirming the non-inferiority hypothesis. Of note, without the adaptation of the study hypothesis, the trial would have been unsuccessful, and a new trial would have been necessary to test non-inferiority. Another trial with an adaptive design was the study by Kapur and colleagues [24]. They investigated the efficacy of three intravenous anticonvulsive agents (levetiracetam, fosphenytoin, and valproate) for the treatment of convulsive status epilepticus that was unresponsive to benzodiazepines. Patients were randomized 1:1:1 into three study groups until 300 people were enrolled. Subsequently, patients were allocated based on response to treatment and interim analyses were scheduled every additional 100 patients enrolled to evaluate early success or futility of the study. Once 400 patients were enrolled, the interim analysis revealed that the predefined criteria of futility were met and the study was stopped by the monitoring board, preventing further treatment of patients with an ineffective drug. In a randomized, double-blind, placebo-controlled trial, Bhatt et al. compared the rate of death, myocardial infarction, ischemia-driven revascularization, and stent thrombosis after percutaneous coronary intervention (PCI) in patients treated with cangrelor or clopidogrel [25]. A sample size of 10,900 patients was estimated to reach 85% statistical power based on a composite primary endpoint forecast of 5.1% in the clopidogrel group and 3.9% in the cangrelor group. However, a scheduled interim analysis was performed when 70% of the patients were randomized, showing small variations in the expected difference in relative risk between the two groups. To avoid a possible loss of statistical power, a new sample size was calculated. Finally, 11,145 patients were enrolled and cangrelor proved to significantly reduce the rate of ischemic events compared with clopidogrel (adjusted odds ratio 0.78; 95% confidence interval [CI], 0.66–0.93, *p* = 0.005). Adaptation of the sample size allowed to reach an adequate statistical power, preventing study failure.

## 6. Where Are We in IBD?

Adaptive designs were recently introduced in the IBD field. The Program of Ulcerative Colitis Research Studies Utilizing an Investigational Treatment—Subcutaneous (PURSUIT-SC) study investigated safety and efficacy of SC golimumab for induction therapy of patients with moderate-to-severe ulcerative colitis (UC) [26]. This study integrated phases IIb and IIIa of a clinical trial, using an adaptive randomization process for phase II (dose-ranging) [26]. In phase II, 169 patients were randomized (1:1:1:1) to receive the placebo or golimumab (100/50, 200/100, or 400/200 mg) at weeks 0 and 2 [26]. At week 6, a higher rate of response and clinical remission (30% or 3 point decrease in the Mayo score compared with baseline and Mayo score ≤ 2 points, with no individual subscore > 1, respectively), mucosal healing (Mayo endoscopic subscore ≤ 1), and improvements of the Inflammatory Bowel Disease Questionnaire (IBDQ) were found in patients treated with golimumab compared with the placebo [26]. Based on clinical, endoscopic, pharmacokinetic, and safety data, golimumab 400/200 mg and golimumab 200/100 mg were selected as the induction regimens for a phase III study [26]. Although the authors defined the study design as an adaptive seamless phase II/III clinical design, no adaptation could be identified in this study. The phase III study did not begin after a phase II interim analysis, but at the end of the phase II study, as occurs in traditional study designs. In addition, a sample size of approximately 750 patients was required to reach a 90% statistical power in the phase III trial, ignoring patients enrolled during the phase II trial. Thus, the advantages in terms of sample size reduction and trial duration shortening expected from a seamless phase II/III design were lost. Similarly, in the PURSUIT maintenance study, an adaptive randomization procedure was described [27]. Patients who responded to the placebo during the induction phase and those who did not respond to the placebo or golimumab were not randomized [27]. On the other hand, patients who lost clinical response after initial placebo treatment were randomized to golimumab 50 mg, patients initially treated with golimumab 50 mg were shifted to golimumab 50 mg or 100 mg every four weeks, and patients initially treated with golimumab 100 mg were re-randomized to golimumab 100 mg or 200 mg every four weeks in case of loss of response. Interestingly, the study arm with golimumab 200 mg was discontinued during the ongoing study and patients treated with this dosage were shifted to golimumab 100 mg. However, no adaptation was detected in patients’ allocations. In fact, the 200 mg dose interruption was due to a protocol amendment based on data from rheumatologic studies rather than determined by data previously accumulated, as expected in an adapted design clinical trial [28,29]. Additional IBD studies with adaptive designs are ongoing and will provide further data on the advantages and disadvantages of this investigative approach (NCT03466411 and NCT03758443).

## 7. Outlook

In recent years, several molecules have been developed and approved for the treatment of IBD patients, but the disease remission rate still remains modest, highlighting the real need for new drugs and for alternative development strategies [30]. IBD clinical trials are increasingly complex and sophisticated, involving new outcomes (e.g., patient-reported outcomes, biomarkers, and histological healing), central reading, and re-randomization [31]. The use of adaptive designs could significantly simplify scientific research focusing on the most relevant data, shortening trials’ duration, and allowing to make the optimal use of available resources and to reduce costs. The benefits of an adaptive approach in the field of IBD are evident and cannot be neglected as it could have a significant impact on both the results and costs of the studies. They could allow to adapt the sample size, ensuring an adequate statistical power or the use of biomarkers (e.g., fecal calprotectin) to monitor the response to therapy and to validate them in clinical practice. Moreover, adaptive designs could help overcome one of the main challenges of IBD trials, which is patient recruitment [32]. In fact, IBD trials require an extremely large number of patients because of their high variability including not only the study population (e.g., patients with stenosing or fistulizing disease, extra-intestinal manifestations, elderly or pediatric subjects), but also available treatments (e.g., immunosuppressants, biological, or small molecules) and evolving study outcomes (e.g., biochemical, endoscopic, histological, and patient-reported outcomes (PRO)) [32]. Study adaptations could reduce the number of patients needed to obtain data with sufficient statistical power and ensure that patients are treated with the best available treatment. Adaptive designs should be considered not only in exploratory trials aimed at the development of new drugs, but also in confirmatory trials proving the efficacy of strategies/therapies in patients with IBD. However, there is little familiarity with these study designs as underlined by the examples above [26,27]. It should be emphasized that a peculiar feature of this design is to accumulate data during the study period. Any predefined adjustments must be exclusively made on the basis of results obtained by analyzing data collected in scheduled time-points. The use of an adaptive design should always be justified within the study protocol, demonstrating the advantages of this design compared with traditional designs, providing guarantees in the control of type I error rates, and appropriately maintaining data integrity and consistency. If properly planned and conducted, the studies with adaptive designs could be useful in identifying effective and safe molecules for the treatment of IBD patients and reducing their long approval time.

## 8. Conclusions

Adaptive designs are promising and attractive designs with significant versatility and flexibility allowing to identify treatment effects or efficacy trends and to modify the investigation according to study results. Adaptive methods should be specified and well defined in the study protocol in order to reduce bias and difficulty in data interpretation. To date, few studies have used adaptive designs in IBD trials. It is desirable that in the coming years, an increasing attention will be paid to the study planning phase, implementing the use of adaptive designs and focusing more on the advantages deriving from this approach.

## Figures and Tables

**Figure 1 jcm-09-02350-f001:**
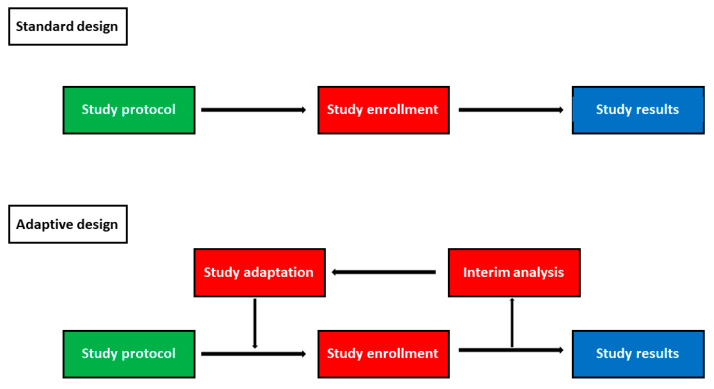
Comparison between standard and adaptive study designs.

**Figure 2 jcm-09-02350-f002:**
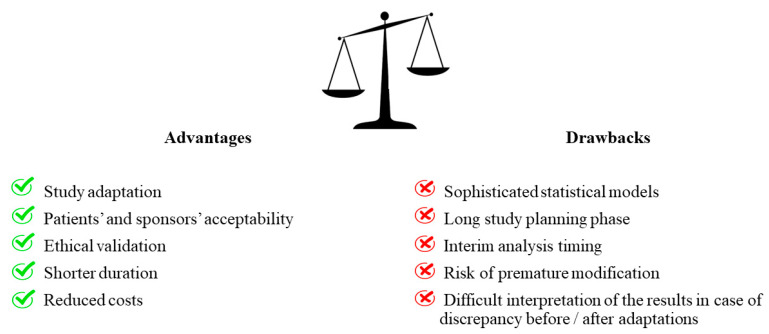
Advantages and drawbacks of adaptive trials.

**Table 1 jcm-09-02350-t001:** Characteristics of adaptive designs.

Adaptive Designs	Main Characteristic
Adaptive dose-finding design	To modify patients’ allocations to identify the optimal drug dosage
Adaptive hypothesis designs	To modify the study hypothesis (e.g., non-inferiority, superiority) or endpoint (primary and/or secondary)
Adaptive group sequential design	To modify dose or treatment duration
Sample size re-estimation design	To re-estimate the sample size
Adaptive treatment-switching design	To switch a patient from a treatment group to another due to loss of response to initial therapy
Adaptive randomization design	To adjust randomization schedules increasing the number of patients randomized to the most beneficial groups
Pick-the-winner/drop-the loser design	To modify the study arms removing less effective ones or adding groups that seem effective
Biomarker adaptive design	To adapt patient allocation based on treatment response to biomarkers
Seamless phase II/III design	To combine phases II and III of clinical trials.
Multiple adaptive design	To combine multiple adaptive designs in the same study

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
