# Peer review of "Adaptive Designs: Lessons for Inflammatory Bowel Disease Trials"

_jcm, 2020, doi:10.3390/jcm9082350_

Round 1
Reviewer 1 Report
This is a summary review of the use of adaptive designs in therapeutic trials. The review does not present new information but seeks to contextualise the use of these techniques for inflammatory bowel disease research. The paper is well-written and allows sometimes complex concepts to be understood by clinicians in this field.
While plagiarism is not detected, fig 1 is very similar to a figure in Pallmann et al. BMC Medicine (2018) 16:29, which is not cited, and - perhaps understandably - several of the examples given in section 5 are also cited in similar review articles (eg Pallmann 2018, Bhatt 2016).
Section 2 may be more clearly presented as a table.
In section 3, the sentence “Adaptive designs have a greater acceptability than traditional studies” should be clarified to identify to whom (trial subjects?) they are more acceptable.
Some of the advantages in IBD that are not mentioned in section 3 might include: avoiding conducting underpowered trials (as has been the case in IBD), identification of patients likely to respond to a particular treatment (relevant in IBD where personalised medicine is of increasing importance)
Section 4 does not really give a historical perspective of the development of adaptive trial design – perhaps this section could be slightly reworked or renamed?
In section 5, it may be better to emphasise the ways in which an adaptive design changed the conduct of each study rather than reporting the study endpoints so thoroughly.
Section 7 is well-argued, but may be improved by more extended discussion of the IBD-related factors (heterogenous diseases, different outcomes, sometimes heterogenous treatments – for example faecal microbial transfer) that make conventional trials unsuccessful and which can be addressed by adaptive design.
Author Response
Reviewer 1
This is a summary review of the use of adaptive designs in therapeutic trials. The review does not present new information but seeks to contextualise the use of these techniques for inflammatory bowel disease research. The paper is well-written and allows sometimes complex concepts to be understood by clinicians in this field.
Reply: We thank the reviewer for his/her positive comment.
While plagiarism is not detected, fig 1 is very similar to a figure in Pallmann et al. BMC Medicine (2018) 16:29, which is not cited, and - perhaps understandably - several of the examples given in section 5 are also cited in similar review articles (eg Pallmann 2018, Bhatt 2016).
Reply: We are impressed with this reviewer comment. However, we confirm that our figure is original. We have tried to summarize the differences between standard design and adaptive design as simply as possible. In addition, we have provided the most representative examples of adaptive designs after long and careful research of the available literature evidence.
Section 2 may be more clearly presented as a table.
Reply: We thank the reviewer for this suggestion. We have added table 1, summarizing the main characterisics of adaptive designs as requested.
In section 3, the sentence “Adaptive designs have a greater acceptability than traditional studies” should be clarified to identify to whom (trial subjects?) they are more acceptable.
Reply: We thank the reviewer for this comment. We have clarified the sentence as follows: “Adaptive designs have greater patient and sponsor acceptability than traditional studies”.
Some of the advantages in IBD that are not mentioned in section 3 might include: avoiding conducting underpowered trials (as has been the case in IBD), identification of patients likely to respond to a particular treatment (relevant in IBD where personalised medicine is of increasing importance)
Reply: We gratefully thank the reviewer for this interesting comment. We have added the suggested advantages in our “Advantages and drawbacks of adaptive designs” section: “In addition, adapting the study design by recalculating the sample size or changing the study arms allows to avoid conducting underpowered studies and to identify the patients who are likely to respond most to a specific treatment.”
Section 4 does not really give a historical perspective of the development of adaptive trial design – perhaps this section could be slightly reworked or renamed?
Reply: We have modified the title of this section as suggested: “Evolution of adaptive designs: the regulatory authorities' point of view”.
In section 5, it may be better to emphasise the ways in which an adaptive design changed the conduct of each study rather than reporting the study endpoints so thoroughly.
Reply: We thank the reviewer for the comment. We believe that careful description of study design and endpoints is essential for understanding the impact of adaptive designs. Of note, as recommended by the reviewer, we further emphasized the benefits of adaptive designs in each example.
Section 7 is well-argued, but may be improved by more extended discussion of the IBD-related factors (heterogenous diseases, different outcomes, sometimes heterogenous treatments – for example faecal microbial transfer) that make conventional trials unsuccessful and which can be addressed by adaptive design.
Reply: We have appreciated the reviewer's comment and further discussed in our “outlook” the impact of adaptive designs in IBD trials: “Moreover, adaptive designs could help overcome one of the main challenges of IBD trials, which is patient recruitment [32]. In fact, IBD trials require an extremely large number of patients because of their high variability including not only the study population (eg patients with stenosing or fistulizing disease, extra-intestinal manifestations, elderly or pediatric subjects), but also available treatments (eg immunosuppressants , biological, or small molecules) and evolving study outcomes (eg biochemical, endoscopic, histological, and patient reported outcomes (PRO)) [32]. Study adaptations could reduce the number of patients needed to obtain data with sufficient statistical power and ensure that patients are treated with the best available treatment”.
Reviewer 2 Report
The manuscript entitled "Adaptive designs: lessons for inflammatory bowel diseases" by Ferdinando D’Amico et al. presents an interesting new study design that should allow a better and faster evaluation of potential new therapeutic drugs in IBD, since the conventional (old) study design is characterized by long study duration and high failure rate.
The manuscript is well-written and the given examples are helpful to understand the sophisticated topic.
The introduction of "adaptive study designs" is important since the is still an urgent need for new therapeutic drugs in IBD.
Author Response
Reviewer 2
The manuscript entitled "Adaptive designs: lessons for inflammatory bowel diseases" by Ferdinando D’Amico et al. presents an interesting new study design that should allow a better and faster evaluation of potential new therapeutic drugs in IBD, since the conventional (old) study design is characterized by long study duration and high failure rate. The manuscript is well-written and the given examples are helpful to understand the sophisticated topic. The introduction of "adaptive study designs" is important since the is still an urgent need for new therapeutic drugs in IBD.
Reply: We gratefully thank the reviewer for her/his positive comment and interest in our article.
Reviewer 3 Report
The manuscript is an excellent summary about the adaptive design in clinical studies, which potentially can optimize clinical studies and reduce the associated time and financial burden.
Adaptive designs, if planned properly offer obvious advantages such as shorter duration and reduced cost, along with the ability to make adaptation to stop the study upon non-inferiority/superiority of the tested drugs. However, a potential drawback was not mentioned in the manuscript: that this type of design may require repeated studies, increasing maximal sample size.
Further, there are other ongoing adaptive studies in IBD (guselmumab, TD-1473, crofelemer) that can be mentioned in this manuscript.
Author Response
Reviewer 3
The manuscript is an excellent summary about the adaptive design in clinical studies, which potentially can optimize clinical studies and reduce the associated time and financial burden.
Reply: We gratefully thank the reviewer for her/his positive comment.
Adaptive designs, if planned properly offer obvious advantages such as shorter duration and reduced cost, along with the ability to make adaptation to stop the study upon non-inferiority/superiority of the tested drugs. However, a potential drawback was not mentioned in the manuscript: that this type of design may require repeated studies, increasing maximal sample size.
Reply: We thank the reviewer for this comment, however we believe that adaptive designs lead to reduced sample size and reduced risk of repeated studies. In fact, as reported in the examples listed in the "Lessons from non-IBD trials" section, the adaptation of the study design allowed in several cases to prevent the study from failing and to prevent a new study from being designed.
Further, there are other ongoing adaptive studies in IBD (guselmumab, TD-1473, crofelemer) that can be mentioned in this manuscript.
Reply: We thank the reviewer for this interesting suggestion. We have mentioned the ongoing studies with adaptive design on guselkumab (NCT03466411) and TD-1473 (NCT03758443) as requested. Conversely, we did not mention studies on crofelemer as we could not identify on clinicaltrial.gov registered IBD trials with adaptive design on IBD patients